# TRI-FACTOR SALIENCY: A LOW-DIMENSIONAL REPRESENTATION FOR EFFICIENT AND DIVERSITY-AWARE VIDEO TOKEN PRUNING

## ABSTRACT

The quadratic computational overhead of self-attention severely limits the application of Large Vision-Language Models (LVLMs) to long-form video. While training-free token pruning offers a promising avenue for acceleration, current methods still struggle for balancing the token diversity and pruning efficiency. Query-based approaches prune tokens irrelevant to a specific prompt, but consequently sacrifice the intrinsic diversity of the video content. Conversely, methods that preserve diversity by clustering or matching based on the raw, high-dimensional token features incur prohibitive computational costs, making them impractical for long video inputs. In this work, we challenge the assumption that preserving diversity necessitates expensive computations in the original high-dimensional feature space. We hypothesize that a low-dimensional yet informative representation engineered for pruning can achieve comparable results with a fraction of the overhead. To validate this, we propose a framework that first projects the original token features into a highly informative 3D "saliency-space." This projection is achieved via our Tri-Factor Saliency (TFS) model, which computes three largely orthogonal sub-features from a local spatio-temporal neighborhood: (1) Dynamic Saliency, which captures the magnitude of movement; (2) Regional Saliency, which identifies coherent objects that stand out from their background; and (3) Focal Saliency, which pinpoints unpredictable, fine-grained details. This low-dimensional representation enables subsequent entity-aware clustering and diversity-preserving stratified sampling to be performed with minimal computational cost. Our experiments show that this approach allows for the pruning of up to 75% of tokens while retaining 95% of the original model's performance on video understanding benchmarks. Our work demonstrates that a well-designed, low-dimensional perceptual projection can effectively replace expensive high-dimensional feature matching for video token pruning, charting a new course that achieves both high efficiency and strong diversity preservation.

## 1 INTRODUCTION

The remarkable advancements of Large Vision-Language Models (LVLMs) have opened new frontiers in multimodal understanding. However, extending their capabilities from static images to the dynamic and lengthy nature of video presents a significant computational challenge. The self-attention mechanism exhibits a computational complexity that scales quadratically with the number of input tokens. For video, where a single minute can correspond to tens of thousands of visual tokens. Consequently, developing effective token reduction strategies is a critical prerequisite for the practical deployment of LVLMs in real-world video applications.

Training-free, plug-and-play methods are highly desirable as they do not require costly retraining or alter the weights of pretrained models. Current approaches in this domain, however, present a difficult trade-off. One major line of work performs pruning based on the relevance of video tokens to a specific text query. While effective for query-focused tasks, this approach inherently compromises the preservation of the video's intrinsic content diversity, making the resulting token set less suitable for general-purpose analysis or open-ended dialogue. Conversely, another line of work aims to preserve diversity by matching or clustering tokens based on their raw, high-dimensional features.

These methods face a different challenge: the computational cost of operating in a high-dimensional space (e.g., D=2048 or more) becomes a significant bottleneck itself, often negating the potential speed-ups for long video inputs.

In this work, we question the prevailing assumption that preserving visual diversity necessitates expensive computations in the original high-dimensional feature space. We hypothesize that a compact, low-dimensional representation, specifically engineered to capture fundamental perceptual cues, can guide an equally effective pruning strategy with only a fraction of the computational overhead. To test this hypothesis, we introduce a novel framework that first projects the original high-dimensional features into a compact and highly informative 3D "saliency-space."

This projection is achieved via our Tri-Factor Saliency (TFS) model, which computes three largely complementary sub-features from a local spatio-temporal neighborhood:

(1) **Dynamic Saliency**: Derived from the magnitude of local motion vectors, this factor isolates regions with dynamic activity.

(2) **Regional Saliency**: Calculated via center-surround feature contrast, this factor identifies coherent objects and areas that are salient relative to their broader context.

(3) **Focal Saliency**: Based on local feature reconstruction error, this factor pinpoints unpredictable, fine-grained details and complex patterns.

This compact 3D representation, augmented with spatio-temporal coordinates, enables highly efficient downstream processing. The TFS features guide a sophisticated pruning pipeline that includes adaptive video segmentation to handle varying scene dynamics, followed by location-aware clustering on the low-dimensional space. Within each identified entity, a stratified sampling mechanism is then employed to explicitly preserve the diversity of the underlying saliency signals. Our primary contributions can be summarized as follows:

(1) We propose the Tri-Factor Saliency (TFS) model, a compact, low-dimensional, and interpretable representation of token importance, designed as an efficient proxy for high-dimensional features in the context of video pruning.

(2) We present a complete, zero-shot pruning framework that leverages the TFS representation to perform efficient, diversity-aware sampling through adaptive segmentation and stratified clustering.

(3) Through extensive experiments, we demonstrate that our approach reduces computational overhead while maintaining high performance on challenging video understanding benchmarks, thereby validating our core hypothesis that a well-designed, low-dimensional projection is sufficient for intelligent video token pruning.

## 2 RELATED WORKS

Our work is situated at the intersection of efficient vision-language models and, more specifically, training-free token reduction techniques for Vision Transformers.

### 2.1 EFFICIENT INFERENCE FOR VISION-LANGUAGE MODELS

The substantial computational requirements of Large Vision-Language Models (LVLMs) (Achiam et al., 2023; Bai et al., 2025; Wang et al., 2025; Li et al., 2024a), driven by the self-attention mechanism's quadratic complexity, have motivated extensive research into efficiency optimizations. Broadly, these efforts include model compression techniques such as quantization and weight pruning (Liang et al., 2021; Zhou et al., 2024b; Qu et al., 2025), as well as the design of efficient architectures incorporating alternatives to full self-attention (Zhang et al., 2025; Sun et al., 2025). Our work complements these approaches by focusing on token reduction, a plug-and-play strategy that decreases the input sequence length for any standard Transformer architecture without requiring modifications to its weights or extensive retraining.

## 2.2 TRAINING-FREE TOKEN REDUCTION

Training-free methods are particularly valuable as they can be applied to off-the-shelf pretrained models. Current approaches in this domain can be primarily categorized by the information source used to guide the reduction.

One line of work is **Query-Aware Pruning** (Cao et al., 2023; Chen et al., 2024; Huang et al., 2024; Chen et al., 2024; Xing et al., 2025; Lin et al., 2025; Arif et al., 2024), which utilizes the text query to assess the importance of each visual token. Tokens with low relevance to the query are discarded. While effective for specific, query-focused tasks, this strategy's reliance on a given prompt makes the resulting token set less suitable for general-purpose video representation, which is often required for open-ended dialogue.

To preserve the intrinsic diversity of the video, **Content-Aware Reduction** methods (Alvar et al., 2025; Li et al., 2025) operate solely on the visual features. This paradigm includes both token pruning and token merging, with the latter being notably represented by ToMe (Bolya et al., 2023), which progressively combines the most feature-similar tokens. A key challenge for all content-aware methods that aim to preserve diversity is the computational cost. To make informed decisions, these approaches often rely on similarity comparisons or clustering within the raw, high-dimensional feature space. This process itself can become a significant overhead for long videos, creating a fundamental trade-off between the quality of the token selection and the efficiency of the reduction algorithm. Therefore, developing a method that can guide a high-fidelity, diversity-aware pruning process without incurring the high cost of global, high-dimensional feature analysis remains a key challenge.

## 3 METHODS

Our proposed framework is a training-free module designed to intelligently prune video tokens for efficient LVLM inference. The entire process is self-contained, operating solely on the feature sequence provided by a pretrained vision encoder. As illustrated in Figure 1, the framework consists of three main stages: (1) the computation of our proposed Tri-Factor Saliency (TFS) representation for every token; (2) an adaptive segmentation of the video into dynamically consistent scenes; and (3) a per-segment, diversity-aware pruning process that leverages clustering and a novel stratified sampling mechanism.

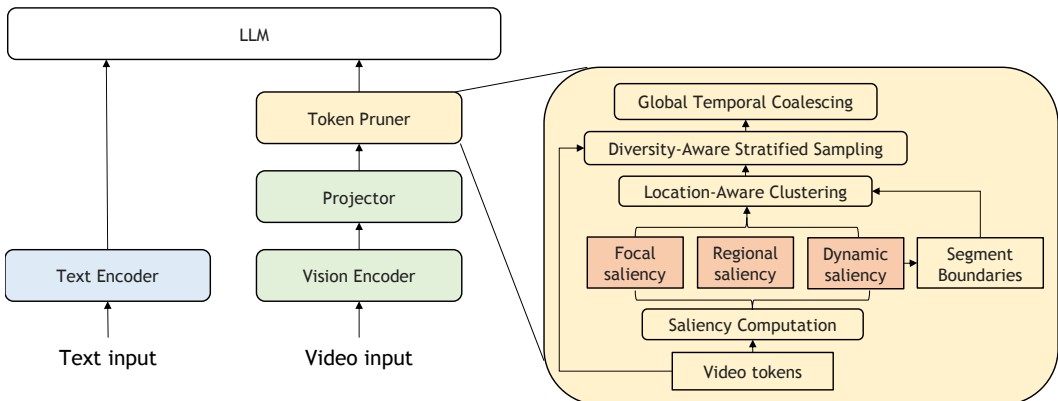

Figure 1: An overview of our proposed pruning framework. The system first computes a 3D saliency representation for all tokens. This representation then guides a two-stage process of adaptive segmentation followed by per-segment, diversity-aware clustering and sampling. After that, a global temporal coalescing process is executed to remove tokens that temporally unchanged.

## 3.1 THE TRI-FACTOR SALIENCY (TFS) REPRESENTATION

We assume that the visual importance of a token, for the purpose of pruning, can be effectively captured by projecting its high-dimensional feature into a compact, 3D representation. Our Tri-Factor Saliency (TFS) model is composed of three complementary perceptual scores. Let $V = \{v_i\}_{i=1}^N \in \mathbb{R}^{N \times D}$ be the input sequence of video tokens, where $N = T \times H \times W$ is the total number of tokens and $D$ is the feature dimension. Each token $v_i$ has spatio-temporal coordinates $(t_i, h_i, w_i)$.

### 3.1.1 DYNAMIC SALIENCY ($S_M$)

Dynamic Saliency aims to quantify the magnitude of motion. For each token $v_i$ at time $t_i > 0$, we first estimate its motion vector $\mathbf{mv}_i \in \mathbb{Z}^2$ by finding the spatial offset $(\Delta h, \Delta w)$ of the most feature-similar token within a spatial neighborhood in the preceding frame $(t_i - 1)$. This feature-based matching serves as a proxy for semantic-level optical flow. The final score for a token $v_i$ is the average magnitude of the motion vectors within its local spatio-temporal neighborhood $\mathcal{N}(i)$:

$$S_M(i) = \frac{1}{|\mathcal{N}_{\text{real}}(i)|} \sum_{j \in \mathcal{N}(i)} ||\mathbf{mv}_j||_2 \tag{1}$$

where $|\mathcal{N}_{\text{real}}(i)|$ is the count of valid neighbors. For $t_i = 0$, $\mathbf{mv}_i$ is defined as $(0,0)$. This aggregation makes the score robust to spurious individual motion estimates and ensures it reflects the saliency of a coherently moving region.

### 3.1.2 REGIONAL SALIENCY ($S_R$)

After accounting for motion, a key aspect of static importance is the presence of coherent regions that stand out from the background. Regional Saliency identifies coherent regions that are salient relative to their larger context. For each token $v_i$, we define a "center" neighborhood $\mathcal{N}_c(i)$ and a larger, non-overlapping "surround" neighborhood $\mathcal{N}_s(i)$. The raw saliency is their cosine distance:

$$S_{R,\text{raw}}(i) = 1 - \frac{\mu_c(i) \cdot \mu_s(i)}{||\mu_c(i)||_2 ||\mu_s(i)||_2}, \tag{2}$$

where $\mu_c(i)$ and $\mu_s(i)$ are the mean feature vectors of the respective neighborhoods. To ensure robustness against statistical instability from few neighbors at frame boundaries, we apply a credibility blending technique. We compute a confidence score $c_R(i)$ based on the number of valid surround neighbors $|\mathcal{N}_{s,\text{real}}(i)|$:

$$c_R(i) = \text{norm}(\log(1 + |\mathcal{N}_{s,\text{real}}(i)|)), \tag{3}$$

where $\text{norm}(\cdot)$ is a min-max normalization across all tokens. The final score is a blend of the raw score and the global median of raw scores, $\tilde{S}_{R,\text{raw}}$:

$$S_R(i) = c_R(i) \cdot S_{R,\text{raw}}(i) + (1 - c_R(i)) \cdot \tilde{S}_{R,\text{raw}}. \tag{4}$$

This smoothly transitions the score to a neutral default value in low-confidence regions.

### 3.1.3 FOCAL SALIENCY ($S_F$)

The third axis of importance relates to information content, as defined by predictability. $S_F$ is designed to identify tokens that are surprising or unpredictable given their immediate context, thus representing high information content. For each token $v_i$, we compute its reconstruction $\hat{v}_i$ from its spatial neighbors $\mathcal{N}_{sp}(i)$ using a similarity-weighted average:

$$\hat{v}_i = \sum_{j \in \mathcal{N}_s(i)} w_j v_j, \quad \text{where} \quad w_j = \frac{\exp(\cos(v_i, v_j))}{\sum_{k \in \mathcal{N}_s(i)} \exp(\cos(v_i, v_k))}. \tag{5}$$

The raw focal saliency $S_{F,\text{raw}}(i)$ is the cosine distance $1 - \cos(v_i, \hat{v}_i)$. We then apply the same credibility blending technique as in Eq. 4, using the count of valid spatial neighbors $|\mathcal{N}_{s,\text{real}}(i)|$ to compute the confidence, yielding the final robust score $S_F(i)$.

## 3.2 ADAPTIVE VIDEO SEGMENTATION

Real-world videos often consist of multiple scenes with vastly different dynamic paces. Applying a single, global pruning strategy is therefore suboptimal. To address this, we first segment the video into dynamically consistent clips by identifying significant changes in its motion profile. This process is guided by a 1D time-series signal, $P(t)$, which we term the "pace signal."

**Pace Signal Formulation.** For each frame $t$, we derive its pace signal $P(t)$ from the previously computed motion saliency map $S_M[t,:,:] \in \mathbb{R}^{H \times W}$. A naive choice for $P(t)$ might be the mean of the scores in the frame. However, the mean is highly sensitive to the proportion of static background tokens; even a small, fast-moving object (a highly salient event) can be washed out when averaged with thousands of near-zero scores from the background. Conversely, using the maximum value is overly sensitive to singular noisy outliers.

To achieve a statistically robust yet sensitive measure of the frame's peak dynamic intensity, we define the pace signal as the 95th percentile of the motion scores within that frame. This allows the signal to be determined by the most active regions while inherently ignoring the vast majority of the static background and the most extreme outliers. The pace signal is defined as:

$$P(t) = \text{quantile}(S_M[t,:,:], 0.95). \tag{6}$$

**Change-Point Detection.** With the pace signal established, we identify segment boundaries by detecting significant change-points. A frame $t$ is marked as a boundary if it satisfies either of two conditions: (1) **Absolute Discontinuity:** Its pace value $P(t)$ exceeds a high, fixed threshold, $T_{abs}$, indicating a definitive break in temporal coherence. (2) **Relative Pace Change:** The rate of change in pace is a statistical outlier. We compute the pace derivative $dP[t] = |P[t] - P[t-1]|$ (with $dP[0] = 0$) and define a relative threshold $T_{rel}$ indicating significant relative changes.

The final set of boundaries, $B$, is the union of the indices found by both rules, along with the start and end of the video.

$$B = \text{sorted}(\text{unique}(\{0, T\} \cup \{t | P[t] > T_{abs}\} \cup \{t | dP[t] > T_{rel}\})). \tag{7}$$

## 3.3 DIVERSITY-AWARE PRUNING STRATEGY

For each identified segment, we apply our clustering and pruning strategy, detailed in Algorithm 1. The strategy is designed to preserve the diversity of important tokens by first identifying coherent entities via clustering and then sampling within them proportionally.

### 3.3.1 LOCATION-AWARE CLUSTERING FOR ENTITY IDENTIFICATION

To identify coherent visual entities, we perform similarity-based clustering with our saliencies. A clustering algorithm operating on saliency scores alone would be location-agnostic, unable to distinguish two identical but spatially separate objects. To resolve this, we construct a 6D feature vector for each token by concatenating its normalized TFS scores and spatio-temporal coordinates $(t, h, w)$. This encourages the formation of clusters that are not only similar in their importance profile but also spatio-temporally contiguous, aligning with the intuitive notion of a visual "entity."

### 3.3.2 STRATIFIED SAMPLING WITHIN CLUSTERS

A simple Top-K sampling based on a fused score can lead to a loss of diversity. To prevent this, we introduce a stratified sampling mechanism formalized in Algorithm 1. Within each cluster, we execute the following steps: (1) We categorize the cluster's member tokens into strata based on their dominant TFS score (i.e., which of the three scores is highest for that token). (2) The cluster's total token budget is then proportionally allocated among these strata based on their relative sizes. (3) Finally, a Top-K sampling is performed within each stratum based on the magnitude of its dominant score. This ensures that all types of salient features that constitute an entity are proportionally represented in the final pruned set.

---

**Algorithm 1** Per-Segment Diversity-Aware Pruning

---

1: **Input:** Segment tokens $V_{seg}$, Scores $S_{M,seg}, S_{R,seg}, S_{F,seg}$, Retain ratio $r_{base}$, Cluster count $k$
2: **Output:** Kept indices for the segment $I_{kept}$
3: $f \leftarrow \text{concat}(S_{M,seg}, S_{R,seg}, S_{F,seg}, \text{coords})$
4: $L \leftarrow \text{KMeans}(f, k)$                                    ▷ Get cluster label map
5: $B_{total} \leftarrow |V_{seg}| \cdot r_{base}$
6: $B_{per\_cluster} \leftarrow \text{ProportionalAllocate}(|L|, B_{total})$
7: $I_{kept} \leftarrow \emptyset$
8: **for** $c \in \{0, \ldots, k-1\}$ **do**
9:      $V_c \leftarrow \{v_i | L_i = c\}$                             ▷ Tokens in current cluster
10:                                          ▷ Stratify by dominant score type
11:      $S_c \leftarrow \text{concat}(S_{M,seg,c}, S_{R,seg,c}, S_{F,seg,c})$
12:      $D_c \leftarrow \text{argmax}(S_c, \dim = 1)$                  ▷ Categorize by dominant score
13:      $Strata \leftarrow \{\text{motion}, \text{region}, \text{texture}\}$
14:      **for** $j \in \text{Strata}$ **do**
15:          $V_j \leftarrow \{v \in V_c | D_c[v] = j\}$             ▷ Tokens in current stratum
16:          $B_{per\_stratum} \leftarrow |V_j|/|V_c| \cdot B_{per\_cluster}[c]$
17:          $I_{kept\_stratum} \leftarrow \text{TopK}(V_j, S_{c,j}, B_{per\_stratum}[j])$
18:          $I_{kept} \leftarrow I_{kept} \cup I_{kept\_stratum}$
19: **return** $I_{kept}$

---

### 3.3.3 GLOBAL TEMPORAL COALESCING

After the per-segment spatial pruning is complete, a final global post-processing step, **Temporal Coalescing**, is applied to remove the last layer of redundancy. This step targets tokens that were kept by the spatial pruner but are perfectly static relative to the previous frame.

Our approach is direct and computationally efficient. We first identify the set of all tokens in the video (for $t > 0$) whose motion vector magnitude is exactly zero, denoted as $V_{static} = \{v_i | \|\mathbf{mv}_i\|_2 = 0, t_i > 0\}$. A token is considered temporally redundant if it was kept by the spatial pruner but is also part of this static set. The set of tokens to be removed is therefore the intersection $I_{prune} = I_{kept} \cap V_{static}$. The final set of kept tokens is the set difference $I_{final} = I_{kept} \setminus I_{prune}$.

This method, while simple, robustly preserves the first appearance of static objects. A static object appearing at $t = 0$ is protected as we do not apply coalescing to the first frame. An object appearing at a scene cut at frame $t > 0$ will exhibit a non-zero motion vector (due to a poor match in the dissimilar previous frame $t - 1$) and will thus be preserved. Only the subsequent, truly redundant instances of the static object (where $\mathbf{mv} = 0$) are targeted for removal. We acknowledge a potential limitation of this approach: a token's motion vector may be zero if its best match is at the same spatial location, even if the features have changed. We posit that such cases are infrequent for salient tokens and accept this as a reasonable trade-off for the efficiency and simplicity of this global redundancy removal step.

## 4 EXPERIMENTS

In this section, we conduct a comprehensive set of experiments to rigorously evaluate our proposed Tri-Factor Saliency (TFS) pruning framework. Our evaluation is designed to demonstrate our method's efficacy, efficiency, and generality across diverse models and benchmarks.

### 4.1 EXPERIMENTAL SETUP

**Models and Datasets.** To demonstrate the general applicability of our approach, we integrate our TFS-Pruner into Qwen2.5-VL (Bai et al., 2025), InternVL-3.5 (Wang et al., 2025) and LLaVA-OneVision (Li et al., 2024a). Our method is evaluated on five video understanding benchmarks: VideoMME(Fu et al., 2024), LongVideoBench(Wu et al., 2024), MLVU(Zhou et al., 2024a), MVBench(Li et al., 2024b), and TempCompass(Liu et al., 2024). We evaluate all models and benchmarks via VLMEvalKit(Duan et al., 2024).

Table 1: Comprehensive performance comparison across multiple LVLMs and benchmarks. Prefill times are averaged on LongVideoBench. The final column shows the average accuracy retained across all benchmarks relative to the Vanilla model.

| Model | Method | Retain (%) | Prefill (s) | VideoMME | VideoMME-sub | LongVideoBench | MLVU | MVBench | TempCompass | Avg (%) |
|---|---|---|---|---|---|---|---|---|---|---|
| Qwen2.5-VL-7B | Vanilla | 100% | 2.11 | 62.0 | 65.0 | 59.8 | 64.3 | 66.0 | 74.4 | 100% |
| | FastV | 50% | 0.82 | 62.0 | 65.0 | 59.8 | 64.4 | 66.0 | 74.4 | 100% |
| | | 25% | 0.50 | 60.4 | 64.1 | 59.8 | 63.7 | 65.8 | 74.1 | 99.2% |
| | Ours | 50% | 0.76 | 61.9 | 65.6 | 58.7 | 63.4 | 65.8 | 76.6 | 100% |
| | | 35% | 0.68 | 61.6 | 65.1 | 58.6 | 63.1 | 66.2 | 76.8 | 99.9% |
| | | 25% | 0.48 | 60.7 | 64.6 | 58.0 | 62.5 | 66.1 | 76.7 | 99.1% |
| | | 10% | 0.28 | 60.0 | 62.8 | 57.4 | 59.6 | 64.6 | 76.5 | 97.1% |
| Qwen2.5-VL-32B | Vanilla | 100% | 8.01 | 65.3 | 70.4 | 60.1 | 66.5 | 65.1 | 77.7 | 100% |
| | FastV | 50% | 2.87 | 64.1 | 69.6 | 59.8 | 65.7 | 65.1 | 77.5 | 99.2% |
| | Ours | 50% | 2.83 | 64.0 | 69.0 | 59.5 | 65.0 | 65.0 | 77.3 | 98.7% |
| InternVL3.5-8B | Vanilla | 100% | 1.09 | 66.1 | 68.5 | 61.6 | 70.4 | 71.9 | 72.6 | 100% |
| | FastV | 50% | 0.32 | 64.5 | 66.3 | 59.5 | 68.6 | 68.7 | 70.6 | 96.9% |
| | | 25% | 0.26 | 62.7 | 64.8 | 57.7 | 66.8 | 67.0 | 68.8 | 94.3% |
| | Ours | 50% | 0.29 | 64.5 | 67.0 | 61.0 | 69.5 | 65.9 | 71.5 | 97.2% |
| | | 35% | 0.26 | 64.7 | 66.7 | 60.2 | 68.6 | 65.4 | 71.3 | 96.6% |
| | | 25% | 0.25 | 63.6 | 65.9 | 60.0 | 68.0 | 65.2 | 71.2 | 95.9% |
| InternVL3.5-14B | Vanilla | 100% | 1.51 | 67.9 | 71.3 | 63.1 | 74.0 | 73.4 | 74.0 | 100% |
| | FastV | 50% | 0.77 | 65.6 | 68.9 | 61.5 | 72.1 | 69.5 | 72.4 | 96.8% |
| | | 25% | 0.63 | 63.9 | 66.7 | 59.2 | 70.3 | 67.3 | 70.5 | 93.9% |
| | Ours | 50% | 0.73 | 66.3 | 69.8 | 61.3 | 73.8 | 65.3 | 73.8 | 96.9% |
| | | 35% | 0.60 | 66.1 | 69.1 | 60.2 | 73.3 | 64.9 | 73.4 | 96.1% |
| | | 25% | 0.59 | 65.0 | 68.6 | 60.2 | 72.8 | 64.8 | 72.8 | 95.4% |
| LLaVA-OneVision | Vanilla | 100% | 0.42 | 58.3 | 62.3 | 57.8 | 67.0 | 56.6 | 70.4 | 100% |
| | FastV | 50% | 0.34 | 58.1 | 62.1 | 57.6 | 66.8 | 56.1 | 69.1 | 99.3% |
| | | 35% | 0.32 | 57.7 | 61.7 | 57.1 | 65.8 | 55.1 | 68.2 | 98.2% |
| | Ours | 50% | 0.31 | 58.0 | 61.8 | 57.2 | 66.2 | 55.0 | 67.5 | 98.3% |
| | | 35% | 0.28 | 57.8 | 62.1 | 57.2 | 65.9 | 54.6 | 67.0 | 98.0% |

**Implementation Details.** All experiments are conducted on 4x NVIDIA H100 80GB GPUs. Our TFS-Pruner is implemented in PyTorch and Triton. To demonstrate robustness, the following key hyperparameters are kept fixed across all models and datasets. For saliency calculation, the neighborhood coordinates of token $v_i$ with coordinates $(t_i, h_i, w_i)$ are set to $\mathcal{N}(i) = \mathcal{N}_c(i) = (t_i \pm 1, h_i \pm 1, w_i \pm 1)$ and $\mathcal{N}_s(i) = (t_i \pm 1, h_i \pm 1, w_i \pm 1) \setminus \mathcal{N}_c(i)$. For adaptive segmentation, $T_{abs}$ is 0.6, and $T_{rel}$ is the 90th percentile of $dP[t]$. For clustering, we use a fixed number of clusters $k = 20$.

## 4.2 MAIN RESULTS

Table 1 presents a comprehensive comparison of our TFS-Pruner against the baselines across a diverse set of LVLMs and benchmarks. The collected data allows for an analysis of our method's performance retention at various pruning ratios and its performance-efficiency trade-off relative to existing work.

The results indicate a high degree of performance preservation for our method. As shown in the final column of Table 1, which reports the average accuracy retained across all benchmarks, our TFS-Pruner at a 50% token retention ratio achieves over 97% of the vanilla models' performance on most tested architectures. This level of fidelity is largely maintained at more aggressive ratios; for instance, Qwen2.5-VL-7B retains 97.1% of its performance when only 10% of the original tokens are kept.

When compared to the FastV(Chen et al., 2024) at similar retention ratios, our method shows competitive or improved results. On the InternVL3.5-14B model at 25% retention, our TFS-Pruner achieves a 95.4% average performance retention, in contrast to FastV's 93.9%. On Qwen2.5-VL-7B at 25%, both methods yield comparable accuracy (98.1% for our method vs. 99.2% for FastV). The reported prefill times in the table confirm a clear trend of increased acceleration with lower retention ratios, reaching up to a 7.5× speed-up for Qwen2.5-VL-7B at the 10% level. These findings suggest that the proposed TFS representation can effectively guide the pruning process to create a sparse token set that preserves a high degree of the original model's capabilities while significantly reducing computational requirements.

## 4.3 ANALYSIS OF COMPUTATIONAL OVERHEAD

We analyze the overhead of our TFS-Pruner and the resulting end-to-end prefill acceleration across a wide spectrum of model sizes in Table 2. For smaller models, the pruner's overhead, constitutes a noticeable fraction of the original prefill time. However, the total processing time still represents a clear improvement over the vanilla model. As we move to larger models like InternVL3.5-38B, the vanilla prefill time becomes the dominant factor. Here, our pruner's modest overhead is dwarfed by the massive savings in the prefill stage, leading to a much higher end-to-end prefill speed-up of [e.g., 2.1×] at 50% retention. This trend clearly demonstrates the excellent scalability and increasing utility of our method for current and future large-scale vision-language models.

Table 2: Overhead and end-to-end prune & prefill speed-up across various model sizes. All timings are averaged on LongVideoBench. Pruned results are shown for our TFS-Pruner at a 50% token retention ratio.

| Model | Vanilla Prefill (s) | Pruner Overhead (s) | Pruned Prefill (s) | E2E New Time (s) | Speed-up |
|---|---|---|---|---|---|
| LLaVA-OneVision (7B) | 0.42 | 0.35 | 0.31 | 0.66 | 0.6× |
| Qwen2.5-VL (7B) | 2.11 | 0.46 | 0.76 | 1.22 | 1.7× |
| Qwen2.5-VL (32B) | 8.01 | 0.46 | 2.83 | 3.29 | 2.4× |
| Qwen2.5-VL (72B) | 17.11 | 0.49 | 6.13 | 6.62 | 2.6× |
| InternVL3.5 (8B) | 1.09 | 0.58 | 0.29 | 0.87 | 1.3× |
| InternVL3.5 (14B) | 1.51 | 0.57 | 0.73 | 1.30 | 1.2× |
| InternVL3.5 (38B) | 3.22 | 0.57 | 1.44 | 2.01 | 1.6× |

## 4.4 COMPATIBILITY WITH FASTV

To investigate the compatibility of TFS-Pruner with FastV, we present a focused analysis on the LongVideoBench benchmark in Table 3, comparing configurations at a similar final retention ratio of approximately 25%.

Table 3: Compatibility analysis with FastV on LongVideoBench.

| Model | Method | Final Retain (%) | LongVideoBench Acc. |
|---|---|---|---|
| Qwen2.5-VL-7B | Vanilla | 100% | 59.8 |
| | FastV (25%) | 25% | 57.7 |
| | Ours (25%) | 25% | 58.0 |
| | Ours (50%) + FastV (50%) | 25% | **58.9** |
| InternVL3.5-14B | Vanilla | 100% | 63.1 |
| | FastV (25%) | 25% | 59.2 |
| | Ours (25%) | 25% | **60.2** |
| | Ours (50%) + FastV (50%) | **25%** | 59.4 |

For Qwen2.5-VL-7B, we observe a clear synergistic effect. The combined approach at a 25% final retention ratio achieves a score of 58.9, outperforming both our standalone pruner (58.0) and the projected score for FastV at the same ratio. This suggests that our perceptual-based pruning provides a cleaner, more condensed input for FastV, allowing its own reduction mechanism to operate more effectively on the most important tokens.

Conversely, for InternVL3.5-14B, we observe a slight interference effect. The combined approach (59.4) underperforms our standalone TFS-Pruner at the same final ratio (60.2). We hypothesize this may occur because our TFS pruning process, while preserving what is perceptually salient, might inadvertently remove tokens that have higher important scores during prefill stage.

In summary, this analysis indicates that while our TFS-Pruner can be effectively combined with orthogonal methods, the interaction is complex and may be model-dependent. This underscores the strength of our TFS-Pruner as a powerful standalone method, and suggests that cascaded pruning strategies are a promising but nuanced avenue for future research.

### 4.5 Qualitative Analysis

To provide an intuitive understanding of our method's decision-making process, we present a series of qualitative results in Figure 2. The visualization showcases our framework's ability to decompose complex scenes into meaningful perceptual components and retain a diverse set of salient tokens.

Dynamic Saliency captures the movement of the person and light changes. Regional Saliency tends to modeling the edge of entities, while it is hard to capture the complex details inside entities (bookcase in the lower left corner), which could be addressed by Focal Saliency. As shown in the **Cluster Map**, our location-aware clustering successfully segments the scene into semantically coherent entities, assigning distinct labels to both dynamic and static entities.

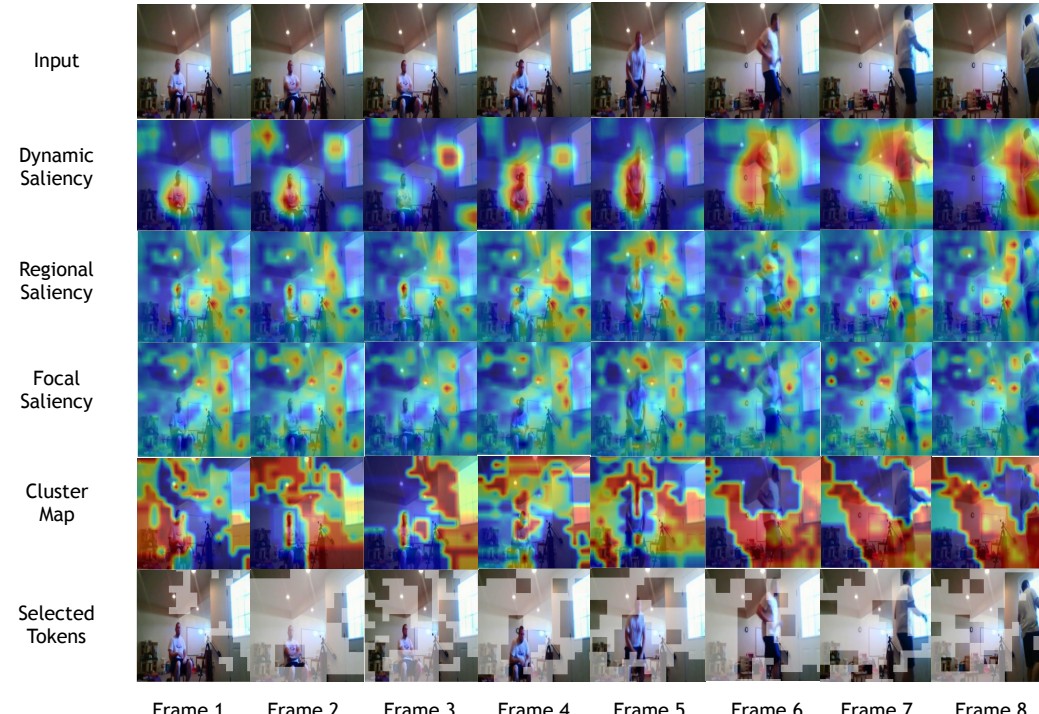

Figure 2: Qualitative results of our method on sample frames. From top to bottom: Input Frame, Dynamic Saliency ($S_M$), Regional Saliency ($S_R$), Focal Saliency ($S_F$), Cluster Map (each color represents a distinct entity), and the final Selected Tokens overlaid on the frame. Our method correctly identifies and preserves a diverse set of tokens corresponding to the most salient static and dynamic entities.

## 5 Conclusion

In this work, we addressed the prohibitive computational cost of applying LVLMs to video by proposing a novel, training-free token pruning framework. Our approach is centered on the Tri-Factor Saliency (TFS) model, which validates our core hypothesis that a compact, low-dimensional representation—decomposed into dynamic, regional, and focal saliency—is sufficient for guiding a sophisticated, diversity-aware pruning strategy. Our extensive experiments demonstrate that this principled method can prune up to 75% of tokens while retaining a high degree of the original model's performance across multiple benchmarks and models. Future work could focus on unifying our task-agnostic perceptual model with task-aware signals or model-specific architectural priors, and extending the pruning framework to an information-preserving token merging paradigm.

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

# A APPENDIX

## A.1 USE OF LARGE LANGUAGE MODELS IN MANUSCRIPT PREPARATION

We acknowledge the use of a large language model (LLM) assistant (Google Gemini) during the preparation of this manuscript. The LLM was utilized for tasks including but not limited to: improving the grammar, clarity, and style of the text; rephrasing and structuring sentences; and assisting in the initial exploration of related literature. We affirm that the core scientific ideas, the design of the proposed algorithms, the execution and analysis of experiments, and the final conclusions presented in this paper are the original contributions of the human authors. The LLM served as a tool for productivity and refinement, and the intellectual ownership of this work rests entirely with the authors.

