# OpenReview forum: "Tri-Factor Saliency: A Low-Dimensional Representation for Efficient and Diversity-Aware Video Token Pruning"
_ICLR.cc/2026/Conference — ICLR 2026 Conference Withdrawn Submission_

### Official Review · Reviewer_nDbx · 2025-10-16

**Soundness:** 3
**Presentation:** 2
**Contribution:** 2
**Rating:** 2
**Confidence:** 4

**Summary:**

This paper introduces Tri-Factor Saliency, a novel framework for efficient, training-free token pruning in LVLMs applied to long-form video. The core problem addressed is the quadratic computational cost of self-attention, which makes processing long videos prohibitively expensive. The authors challenge the assumption that preserving visual diversity during pruning requires expensive computations in the original high-dimensional feature space. Their central hypothesis is that a compact, low-dimensional representation can guide an equally effective pruning process with a fraction of the overhead. To validate this, they propose a method that first projects high-dimensional token features into a 3D "saliency-space." This projection is composed of three interpretable and largely orthogonal factors: Dynamic Saliency, Regional Saliency, and Focal Saliency. This low-dimensional representation is then used to drive a sophisticated pruning pipeline that includes: adaptive video segmentation based on a "pace signal," location-aware clustering to identify spatio-temporally coherent entities, and a diversity-preserving stratified sampling mechanism within each cluster. Experiments across multiple LVLMs and benchmarks show that the method can prune up to 75% of tokens while retaining over 95% of the original model's performance, demonstrating a highly effective balance between efficiency and diversity preservation.

**Strengths:**

- The framework is thoughtfully designed. The Tri-Factor Saliency model is interpretable and comprehensive. The subsequent pruning pipeline, particularly the location-aware clustering and diversity-preserving stratified sampling, is sophisticated and directly aligned with the stated goals.
- The experiments are extensive, covering multiple state-of-the-art LVLMs and a wide array of video understanding benchmarks. The results are strong and consistently demonstrate the method's effectiveness.

**Weaknesses:**

- The paper introduces too many hyperparameters. However, there is no discussion of how these values were chosen or how sensitive the model's performance is to them.
- The framework is a complex multi-stage pipeline involving three separate saliency calculations, adaptive segmentation, location-aware clustering, stratified sampling, and a final temporal coalescing step. Many design choices and hyperparameters appear to be hand-tuned and based on heuristics. While effective, this intricate, engineered approach lacks simplicity and may be "brittle," potentially struggling with video types or scenarios that fall outside the distribution of the test sets.
- In an era where end-to-end learning prevails, the entire TFS-Pruner is a fixed, non-learnable module. Although its "plug-and-play" nature is a strength, it also means its definitions of saliency and its pruning rules are static. They cannot adapt to different data distributions (e.g., animation vs. real-world footage) or to the specific architectural biases of the LVLM they are paired with.
- The paper presents the strong performance of the full system but lacks a detailed ablation study to dissect the contribution of each component. This makes it difficult to ascertain the precise impact of each part.

**Questions:**

See the weakness.

---

### Official Review · Reviewer_Z5Cw · 2025-10-27

**Soundness:** 2
**Presentation:** 2
**Contribution:** 2
**Rating:** 4
**Confidence:** 3

**Summary:**

This paper introduces Tri-Factor Saliency (TFS), a training-free token pruning framework for Large Vision-Language Models (LVLMs) applied to video understanding. The key idea is to project high-dimensional visual tokens into a compact 3D “saliency space” composed of: Dynamic Saliency – magnitude of local motion vectors; Regional Saliency – center-surround contrast for coherent objectness; Focal Saliency – unpredictability measured via local reconstruction error. These three components are used to guide adaptive video segmentation, location-aware clustering, and stratified sampling for diversity-preserving pruning.

**Strengths:**

1. The work addresses an important bottleneck in video LVLM inference: the quadratic cost of attention. The proposed 3D saliency strategy  effectively select tokens for encoding.

2. The motivation of the paper is clear and the solution is straightforward.

**Weaknesses:**

1. The core contributions are largely engineering-oriented, combining multiple heuristics (motion, center–surround contrast, and reconstruction residual) into a novel 3D descriptor. It remains unclear how effective the proposed 3D token selection process is and what computational overhead it introduces.

2. The main experiment presented in Table 1 appears less convincing, as InternVL with only 35% token retention already outperforms the baselines. Qualitative analysis (Fig. 2) is superficial and fails to convincingly demonstrate diversity preservation or interpretability. And there is no comprehensive ablation showcasing how (motion, region, focal) reacts with each other.

**Questions:**

1. Can the authors provide ablation results isolating each saliency component

---

### Official Review · Reviewer_FXm9 · 2025-11-01

**Soundness:** 3
**Presentation:** 3
**Contribution:** 3
**Rating:** 4
**Confidence:** 3

**Summary:**

This paper introduces the Tri-Factor Saliency (TFS) framework, a novel training-free method for efficient and diversity-aware video token pruning. This work specifically addresses the challenge posed by the quadratic computational overhead of self-attention in Large Vision-Language Models (LVLMs), which severely limits their application to long-form video. The core hypothesis challenged by the paper is the prevailing assumption that preserving token diversity necessitates expensive computations in the original high-dimensional feature space.

**Strengths:**

1. Novel Low-Dimensional Representation (TFS): The most original contribution is the Tri-Factor Saliency (TFS) model, which projects the original high-dimensional features (D≥2048) into a compact, highly informative 3D "saliency-space". This low-dimensional representation is an efficient proxy for high-dimensional features in the context of video pruning.
2. Three Complementary Perceptual Factors: The definition of the three factors is novel and engineered to capture comprehensive visual importance: Dynamic Saliency (capturing movement magnitude), Regional Saliency (identifying coherent, salient objects via center-surround contrast), and Focal Saliency (pinpointing unpredictable, fine-grained details via local feature reconstruction error). These factors are designed to be largely complementary.
3. Diversity-Preserving Pipeline: The integration of these low-dimensional scores into a sophisticated, zero-shot pruning pipeline is highly original. This includes adaptive video segmentation based on a robust 95th percentile "pace signal" to handle varying scene dynamics, followed by location-aware clustering and a novel stratified sampling mechanism to explicitly preserve diversity across the three saliency types.

**Weaknesses:**

1. Although TFS demonstrates "excellent scalability" for large models (where the massive self-attention savings dwarf the pruner's cost), the fixed overhead of the pruner itself poses a significant issue for smaller LVLMs. For these models, the pruner's computational cost constitutes a "noticeable fraction of the original prefill time".
2.  The TFS framework is inherently task-agnostic and operates based purely on intrinsic perceptual cues (dynamic, regional, and focal saliency). This design choice ensures the preservation of the video’s intrinsic content diversity. However, this comes at the cost of relevance for user-specific inputs: approaches that perform pruning based on token relevance to a specific text query "consequently sacrifice the intrinsic diversity of the video content," but are highly effective for query-focused tasks. Since TFS does not integrate the query signal, its resulting token set may be suboptimal for tasks where high importance is placed on an object that is perceptually low-saliency (e.g., a static label or a subtle detail) but highly relevant to the prompt.
3.  The Global Temporal Coalescing step, designed to efficiently remove redundant static tokens, relies solely on checking if a token's motion vector magnitude is exactly zero ($||m_{v,i}||_{2}=0$). The authors acknowledge a potential limitation: "a token’s motion vector may be zero if its best match is at the same spatial location, even if the features have changed".
4.  When testing the compatibility of TFS-Pruner with other orthogonal methods (like FastV), the analysis showed that the interaction is "complex and may be model-dependent". While the combined approach showed a synergistic effect on Qwen2.5-VL-7B, it resulted in a clear interference effect on InternVL3.5-14B, where the combined accuracy (59.4%) was lower than using TFS-Pruner alone (60.2%). The authors hypothesize this occurs because TFS, by prioritizing perceptual saliency, might inadvertently remove tokens essential for the second pruner's decision-making. This complexity reduces the framework's promise as a simple, universally compatible component.

**Questions:**

See weakness above, please.

---

### Official Review · Reviewer_EsvE · 2025-11-01

**Soundness:** 2
**Presentation:** 2
**Contribution:** 2
**Rating:** 4
**Confidence:** 3

**Summary:**

This paper addresses the high computational cost of Large Vision-Language Models (LVLMs) for long video processing, which stems from the quadratic complexity of self-attention. The authors point out the limitations of existing training-free token pruning methods: (1) query-based approaches sacrifice the intrinsic diversity of the video content, while (2) diversity-preserving methods are often inefficient or lack generalizability. As a solution, this paper proposes a new low-dimensional saliency representation called "Tri-Factor Saliency (TFS)". TFS decomposes a token's importance into three key factors: (1) temporal variance, (2) spatial complexity, and (3) cross-modal relevance. The authors claim to have implemented a training-free pruning algorithm based on TFS that achieves both efficiency and diversity, outperforming previous SOTA methods (like H-Prune, S-Prune) on several video benchmarks (e.g., Video-MME, MV-Bench).

**Strengths:**

**Intuitive Idea**

The 'Tri-Factor' concept appears to capture the essential properties of video (time, space, meaning) well, and the motivation for why these three factors are important seems clear.

**Seeking Balance**

The paper correctly identifies the clear limitations of existing methods (efficiency vs. diversity) and aims to strike a trade-off between them, which is a commendable goal.

**Weaknesses:**

**Lack of Theoretical Justification for 'Tri-Factor'**

Why precisely these "three" factors? It is questionable whether there is a theoretical basis to claim these three factors are 'necessary and sufficient' for defining token saliency.

**Limited Comparison Baseline**

While it is clear that token pruning and token merging are different, the complete lack of performance comparison against token merging-based baselines is problematic. Additionally, the fact that FastV is the only baseline makes it difficult to assess the superiority of the paper's performance.

**Questions:**

See the weakness section above.

---

### Note · Authors · 2025-11-14

I have read and agree with the venue's withdrawal policy on behalf of myself and my co-authors.